# Sugar-Linked Diethyldithiocarbamate Derivatives: A Novel Class of Anticancer Agents

**DOI:** 10.3390/ijms26125589

**Published:** 2025-06-11

**Authors:** Mohammad Najlah, Niamh McCallum, Ana Maria Pereira, Dan Alves, Niusha Ansari-Fard, Sahrish Rehmani, Ayşe Kaya

**Affiliations:** Pharmaceutical Research Group, School of Allied Health, Faculty of Health, Education, Medicine and Social Care, Anglia Ruskin University, Bishops Hall Lane, Chelmsford CM1 1SQ, UK

**Keywords:** disulfiram, diethyldithiocarbamate (DDC), copper complex, cancer prodrugs, saccharide conjugates, metal-activated therapy, thioglycosidic bond, cancer

## Abstract

Disulfiram (DSF), a well-known anti-alcoholism drug, exhibits potent anticancer activity via its metabolite, diethyldithiocarbamate (DDC), which forms a cytotoxic copper complex that selectively targets cancer stem cells. However, its clinical utility is limited by poor solubility and rapid plasma metabolism. This study explores saccharide-linked DDCs as novel prodrugs designed to enhance stability, solubility, and tumour-selective activation. These compounds feature thioglycosidic bonds that shield the DDC moiety from premature degradation while retaining its metal-chelating function to form the active copper(II)bis(*N*,*N*-diethyldithiocarbamate) (Cu(DDC)_2_) complex. The synthesised derivatives were characterised and evaluated for serum stability and in vitro cytotoxicity across several cancer cell lines, including colorectal, breast, lung, and brain cancers. Copper-complexed saccharide-DDC prodrugs demonstrated remarkable cytotoxicity, with improved biostability and solubility profiles. These findings highlight the potential of saccharide-linked DDCs as stable, copper-activated prodrugs for cancer therapy. Further in vivo studies are warranted to validate their pharmacokinetics and clinical relevance.

## 1. Introduction

Cancer remains one of the most pressing global health challenges, responsible for millions of deaths annually. Despite significant advancements in conventional treatments such as chemotherapy, radiation, immunotherapy, and targeted therapies, many patients still experience treatment failure due to drug resistance, systemic toxicity, and poor selectivity toward cancer cells [1,2].

Drug repurposing has emerged as a promising strategy to expedite the development of such agents by leveraging existing drugs with well-characterised safety profiles [3,4]. Among these, disulfiram (DSF), an FDA-approved drug primarily used to treat alcoholism, has garnered interest for its potent anticancer properties.

DSF exerts its anticancer effects through its metabolite diethyldithiocarbamate (DDC), which chelates metal ions, particularly copper (Cu^2+^), to form a copper(II)bis(*N*,*N-*diethyldithiocarbamate) (Cu(DDC)_2_) complex, a highly cytotoxic agent that inhibits aldehyde dehydrogenase (ALDH), proteasome function, and nuclear factor kappa B (NF-κB) signalling, leading to cancer cell apoptosis [5,6]. However, clinical translation of DSF has been largely unsuccessful due to its poor solubility, instability in plasma, and rapid metabolism, which significantly limit its bioavailability and therapeutic effectiveness [7,8]. DDC itself is inactivated by albumin, reducing its ability to form the active copper complex, thereby necessitating alternative strategies to enhance stability and targeted delivery [9]. Recent reviews have summarised these limitations and outlined the anticancer mechanisms of DSF, highlighting the need for improved delivery strategies and tumour-specific activation approaches [9,10].

Cancer cell selectivity remains a major therapeutic challenge. Prodrugs that exploit tumour-specific features—such as elevated copper concentrations or increased metabolic demands—offer a potential route to enhance selectivity and reduce systemic toxicity. Controlled release of active species from prodrugs at the tumour site can maximise local therapeutic efficacy while minimising off-target damage [6,9,11].

A promising approach to address these limitations involves the development of saccharide-linked DDCs, designed as prodrugs that exhibit enhanced stability in systemic circulation and are selectively activated at tumour sites. The saccharide-linked DDCs evaluated in this study were designed to remain stable in circulation and become activated only under conditions present in the tumour microenvironment, such as elevated Cu^2+^ levels [5,6]. These compounds feature thioglycosidic bonds, whereby the saccharide moiety shields the thiol group on the DDC core from premature metabolic inactivation, while preserving its metal-chelating capacity to form the cytotoxic Cu(DDC)_2_ complex in the tumour microenvironment [12,13]. Although saccharide conjugation has been utilised in various drug delivery platforms to improve water solubility, stability, and bioavailability [14,15], its application in metal-dependent anticancer therapeutics remains underexplored [16].

Several studies have highlighted the role of saccharides in cancer treatment [15]. For example, 2-deoxy-D-glucose (2-DG), a glucose analogue, has been investigated for its ability to target cancer cell metabolism, exploiting the increased glucose uptake of the cells [17]. Similarly, chitosan-derived sugars have been reported to exhibit enhanced antitumour properties when conjugated with metal-based drugs. However, no studies have systematically investigated saccharide-DDC conjugates in combination with metal ions, despite their potential advantages in improving drug stability, selectivity, and bioavailability [18].

This study investigates saccharide-linked DDCs (Figure 1) as a novel class of prodrugs for metal-dependent anticancer therapy. Specifically, we assess their serum stability, cytotoxicity across multiple cancer cell lines, and their capacity to form the active Cu(DDC)_2_ complex. The results offer valuable insights into the rational design of more stable and efficacious metal-chelating therapeutics, with potential clinical relevance in the treatment of colorectal, breast, lung, and brain cancers.

## 2. Results and Discussion

One of the primary challenges of using DSF and its major metabolite, DDC, in cancer therapy is their instability in plasma, leading to rapid metabolism and poor bioavailability. In this study, saccharide-linked DDCs were synthesised and evaluated for their stability in biological environments. The synthesis involved the coupling of DDC moieties with various saccharides via thioglycosidic bonds, using a protection-free, aqueous-phase method adapted from established glycosylation protocols [13]. This strategy enabled efficient conjugation under mild conditions, preserving the functional integrity of both the saccharide and the metal-chelating thiol groups. The resulting prodrugs were structurally characterised and screened for their ability to resist enzymatic degradation and maintain biological activity in serum-containing media.

The saccharide-linked DDC derivatives demonstrated excellent aqueous solubility (>100 mg/mL), which supports the rationale for their design as prodrugs. While this high solubility made precise quantification difficult using standard methods, it represents a practical advantage for formulation and delivery.

Stability tests in foetal horse serum demonstrated that saccharide-linked DDC, particularly *N*,*N*-diethyl-D-glucopyranosyl dithiocarbamate (G-DDC), *N*,*N*-diethyl 2-deoxy-D-glucopyranosyl dithiocarbamate (DG-DDC), and *N*,*N*-diethyl 2-acetoamido-2-deoxy-D-glucopyranosyl dithiocarbamate (NAG-DDC), exhibited significantly improved stability compared to free DSF (Figure 2). This increased stability is attributed to the incorporation of thioglycosidic bonds, which effectively shield the reactive thiol group of the DDC moiety from premature enzymatic degradation and albumin-mediated inactivation in plasma [7,9]. In contrast, DSF is known to be chemically unstable and rapidly metabolized in systemic circulation, leading to reduced formation of the active copper(II) complex, Cu(DDC)_2_, which is essential for its anticancer effects [6,8]. Enhanced serum stability not only prolongs the systemic half-life of these prodrugs but also enables controlled release and tumour-selective activation upon exposure to elevated copper ion concentrations found in malignant tissues. The subsequent formation of the Cu(DDC)_2_ complex results in potent cytotoxic activity by inhibiting the ALDH and proteasome pathways, leading to selective apoptosis in cancer cells [5,6]. Additionally, the saccharide moiety enhances aqueous solubility and facilitates better pharmacokinetics and drug delivery, which are essential for improving therapeutic index and minimizing off-target toxicity [14,15]. Collectively, these features suggest that saccharide-linked DDCs offer a multifunctional platform that addresses the limitations of DSF while enhancing anticancer potency and bioavailability.

The reaction between saccharide-linked DDCs and Cu^2+^ (from CuCl_2_) is a crucial step in the activation of these prodrugs, leading to the formation of the active Cu(DDC)_2_ complex responsible for their anticancer effects. Figure 3 demonstrates that saccharide-linked DDCs undergo rapid cleavage in the presence of Cu^2+^, releasing the DDC ligand, which immediately chelates the copper ion to form a stable Cu(DDC)_2_ complex. This process occurs more efficiently and rapidly compared to DSF in aqueous solution, indicating an improved solubility profile and reactivity in aqueous environments. The higher solubility of saccharide-linked DDCs facilitates their dissolution and interaction with CuCl_2_, ensuring efficient metal coordination. Additionally, microscopy images revealed enhanced cytotoxicity in cancer cells treated with saccharide-linked DDCs in combination with CuCl_2_, further confirming the successful formation of the active complex (Appendix A). All studied derivatives demonstrated similar stability in human serum and reactivity with copper ions; however, we have presented a subset of representative compounds based on preliminary activity and structural diversity to illustrate the general behaviour of the series. Broader cytotoxicity screening is ongoing to support compound prioritisation for further evaluation.

To assess the therapeutic potential of saccharide-linked diethyldithiocarbamate, 3-[4,5-dimethylthiazol-2-yl]-2,5-diphenyl tetrazolium bromide (MTT) cytotoxicity assays were conducted on multiple cancer cell lines, including colorectal (H630, H630 R10), breast (MDA-MB-231), and lung (A549) cancers. The results demonstrated that DG-DDC and G-DDC, in the presence of copper (Cu^2+^), exhibited potent cytotoxic effects, with IC_50_ values lower than 10 µM (Table 1). The cytotoxic effect was most pronounced in colorectal and breast cancer cells, where DG-DDC + Cu^2+^ achieved IC_50_ values in the low micromolar range (5.2 µM for H630 WT and 3.62 µM for MDA-MB-231). Notably, DG-DDC + Cu^2+^ also demonstrated significant activity against the drug-resistant colorectal cancer cell line H630 R10, achieving an even lower IC_50_ value of 5.3 µM. This finding is particularly important, as resistance to standard chemotherapies remains a major obstacle in cancer treatment. The microscopy analysis further confirmed the cytotoxic impact, as cells treated with DG-DDC + Cu^2+^ exhibited characteristic apoptotic features such as membrane blebbing, chromatin condensation, and cell detachment (Appendix A). These findings strongly support the hypothesis that saccharide-linked diethyldithiocarbamates enhance copper-mediated cytotoxicity and may serve as promising candidates for the treatment of both sensitive and drug-resistant cancers.

While copper has been the primary metal of interest due to its role in enhancing dithiocarbamate-mediated cytotoxicity and the relatively high intracellular concentration of copper in cancer cells, other metal ions, such as zinc (Zn^2+^), were also tested for comparison (Figure 4). The results indicate that Zn(DDC)_2_ complexes exhibit less potent cytotoxic activity than their copper counterparts, suggesting that copper is the preferred cofactor for maximizing anticancer activity (Figure 4). This finding is consistent with previous studies demonstrating that Cu(DDC)_2_ complexes induce apoptosis more effectively than Zn(DDC)_2_, likely due to differences in their redox activity, proteasome inhibition capacity, and generation of reactive oxygen species [6,11]. Nevertheless, Zn(DDC)_2_ complexes still displayed significant cytotoxic effects, indicating that zinc remains a valuable alternative metal ion, particularly in contexts where copper regulation is therapeutically challenging or where zinc-mediated pathways could provide complementary anticancer mechanisms [11]. Thus, zinc coordination could offer a secondary strategy to expand the versatility and therapeutic window of saccharide-linked DDC prodrugs.

## 3. Materials and Methods

### 3.1. Methods for the Preparation of Saccharide-Diethyldithiocarbamate (Saccharide-DDC)

Saccharide-linked DDC complexes were synthesized and characterised as previously described [13]. Briefly, unprotected sugars were reacted with sodium DDC in the presence of 2-chloro-1,3-dimethylimidazolinium chloride (DMC) and triethylamine (Et_3_N) as a base. The reaction was performed in a water/acetonitrile mixture at low temperature (−15 °C to 0 °C) for 1 h. The resulting saccharide-DDC products were purified via silica gel column chromatography characterised by ^1^H NMR. All saccharide derivatives with diethyl and/or dimethyl substitution have been previously reported in the literature [13]. The compound XY-DDC, however, is novel and has been highlighted as such, with full characterisation data (^1^H NMR and MS) added to the Appendix A (Appendix A).

### 3.2. In Vitro Biostability

A 0.5 mL aqueous solution of the saccharide-DDC derivative (1 mmol) was preheated at 37 °C, added to 2 mL of horse serum (preheated at 37 °C), and incubated in a shaking water bath at 37 °C (Grant OLS Aqua Pro, Shepreth, UK) and 100 rpm. For free DSF, 25 μL of 4 mg/mL DS in DMSO was pipetted into 2 mL of horse serum diluted with 475 μL of distilled water (preheated at 37 °C). At specific time intervals, aliquots of 200 μL were added to 500 μL of acetonitrile and vortexed for 10 s. The mixture solution was centrifuged at 10,000× *g* for 10 min (Heraeus Fresco 17, Thermo Fisher Scientific, Loughborough, UK). The supernatant “A” was collected, and the pellet was re-suspended in a 0.5 methanol vortex for 30 s and heated in the water bath at 37 °C for 2 min, then vortexed for 10 s again and centrifuged at 10,000× *g* for 10 min to give the supernatant “B”. The supernatant “B” was collected, added to supernatant “A”, and analysed by HPLC. For the control, the same concentrations and previous steps were followed, but by replacing the horse serum with distilled water. The control was used as 100% for the stability calculations. HPLC methods: UltiMate 3000 UHPLC (Thermo Fisher Scientific, Loughborough, UK) with Accucore150-C18 column, 2.1 × 100 mm (Thermo Fisher Scientific, Loughborough, UK) was used. The mobile phase comprised 90% HPLC-grade ACN and 10% HPLC-grade zinc sulphate buffer (10^−10^ mol/L). The flow rate was 0.1 mL/min, and UV detection was performed at 260 nm with an injection volume of 10 μL.

### 3.3. MTT Assay

The cells were seeded in 96-well plates at seeding density of 1 × 10^4^ cells/well in Dulbecco’s modified Eagle’s medium (DMEM) with 10% FBS, 1 mM sodium pyruvate, 2 mM L-glutamine, and 0.1 mM non-essential amino acids. Cells were incubated at 37 °C, 5% CO_2_, and 95% relative humidity. Cells were constantly exposed to different concentrations of the tested compound in combination with 10 μM of CuCl_2_ for 72 h and then subjected to a standard (MTT) assay. The experiments were carried out in triplicate and the IC_50_ values were calculated. For experiments assessing the effect of zinc, ZnCl_2_ was added to the medium at a final concentration of 10 µM.

### 3.4. Reaction with Copper

The reaction was initiated by adding an aqueous solution of the saccharide-DDC derivative (1 mM) to a CuCl_2_ solution (1 mM). At defined time intervals, an aliquot of 50 µL was withdrawn and transferred to Spin X centrifuge tube with a cellulose acetate filter (0.45 µm) containing 450 µL of HPLC water and centrifuged for 10 min at 13,000 rpm. Samples were then analysed by HPLC as described in Section 3.2.

## 4. Conclusions

The results of this study provide compelling evidence that saccharide-linked diethyldithiocarbamates represent a promising class of metal-activated prodrugs with enhanced stability and bioavailability compared to traditional DSF-based therapies. The ability to target cancer stem cells suggests potential applications in combination therapy regimens. Future work should focus on in vivo validation, including pharmacokinetic studies and animal models, to determine the full therapeutic potential of these compounds. Additionally, exploring different saccharide modifications may further optimise drug delivery and antitumour properties. Future studies will include evaluation of the saccharide-linked DDC derivatives in non-cancerous cell lines to assess their tumour selectivity and further characterise their therapeutic potential.

## 5. Patents

Najlah, M. (2023). Saccharide-linked diethyldithiocarbamate prodrugs and their use in cancer therapy (Patent No. WO2023232904A1). Retrieved from https://patents.google.com/patent/WO2023232904A1/en (accessed on 2 April 2025).

## Figures and Tables

**Figure 1 ijms-26-05589-f001:**
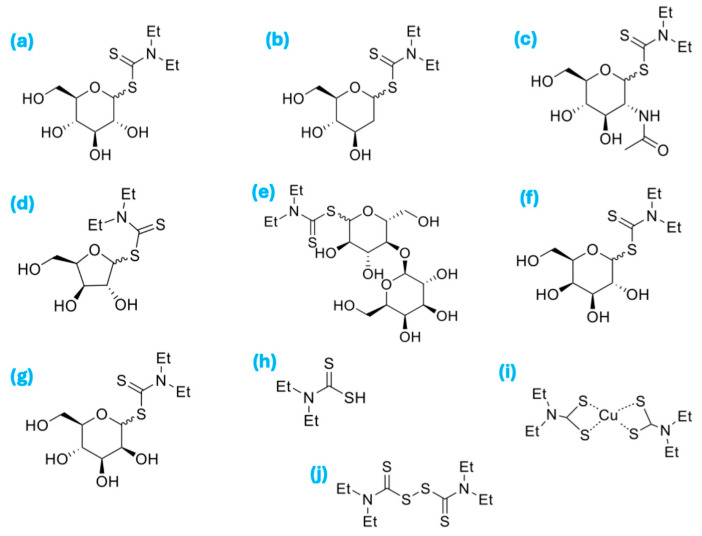
(**a**) *N*,*N*-diethyl-D-glucopyranosyl dithiocarbamate (G-DDC); (**b**) *N*,*N*-diethyl 2-deoxy-D-glucopyranosyl dithiocarbamate (DG-DDC); (**c**) *N*,*N*-diethyl 2-acetoamido-2-deoxy-D-glucopyranosyl dithiocarbamate (NAG-DDC); (**d**) *N*,*N*-diethyl-D-xylopyranosyl dithiocarbamate (XY-DDC); (**e**) *N*,*N*-diethyl-lactosyl dithiocarbamate (La-DDC); (**f**) *N*,*N*-diethyl-D-galactopyranosyl dithiocarbamate (Ga-DDC); (**g**) *N*,*N*-diethyl-D-mannopyranosyl dithiocarbamate (Ma-DDC); (**h**) diethyldithiocarbamic acid (DDC); (**i**) copper(II)bis(*N*,*N*-diethyldithiocarbamate) (Cu-(DDC)_2_); (**j**) disulfiram (DSF).

**Figure 2 ijms-26-05589-f002:**
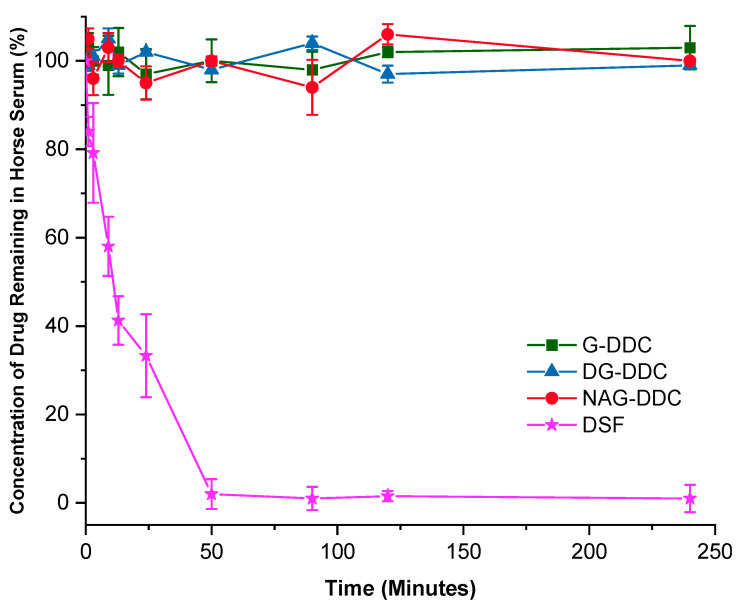
The stability of G-DDC, DG-DDC, and NAG-DDC compared to DSF in foetal horse serum as measured by remaining % over time (min).

**Figure 3 ijms-26-05589-f003:**
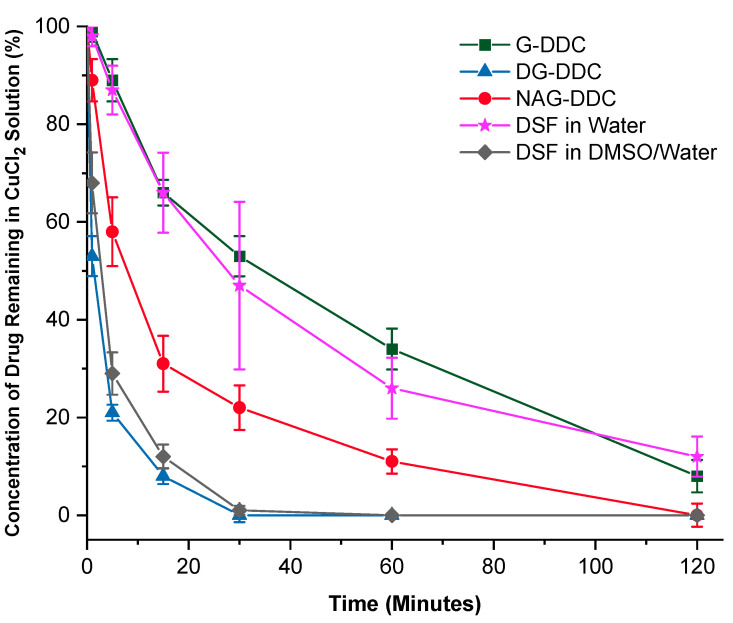
The reaction progress of G-DDC, DG-DDC, NAG-DDC, and DSF in water, and DSF in 50:50 DMSO: water with CuCl_2_ to produce the active Cu(DDC)_2_.

**Figure 4 ijms-26-05589-f004:**
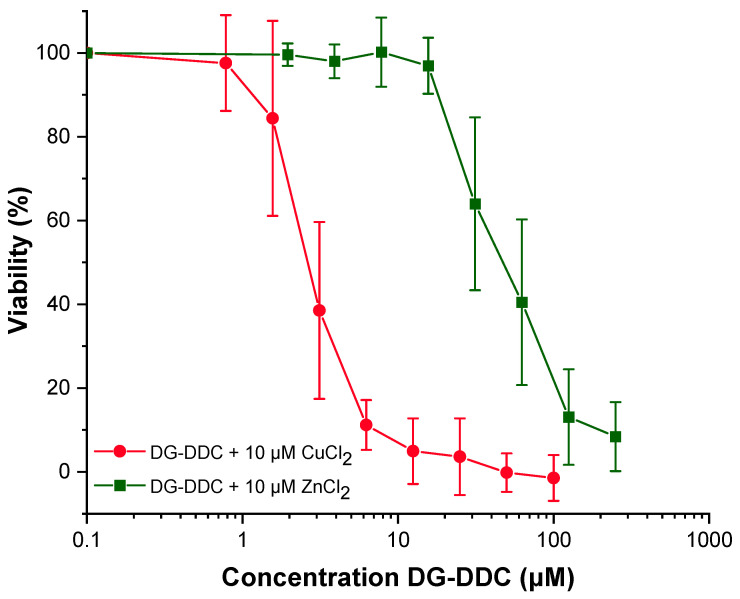
MTT cytotoxicity assay of lung cancer cell line A549 with increasing concentrations of 2-deoxy-glycosyl diethyldithiocarbamate (DG-DDC) with Zn^2+^ and Cu^2+^ (10 µM).

**Table 1 ijms-26-05589-t001:** IC_50_ values for different cancer cell lines.

Compound	H630 WT	H630 R10	MDA-MB-231	A549
G-DDC (µM)	- *	-	2178 ± 356.1	-
G-DDC + Cu^2+^ (µM)	127 ± 13.2	-	347.5 ± 44.7	95.9 ± 21.3
DG-DDC (µM)	-	-	2055 ± 54.70	
DG-DDC + Cu^2+^ (µM)	5.2 ± 1.7	5.3 ± 0.9	3.62 ± 0.22	2.79 ± 0.25
DG-DDC + Zn^2+^ (µM)	-	-	-	49.6 ± 18.6
XY-DDC + Cu^2+^ (µM)	6.9 ± 2.6	-	-	-
NAG-DDC + Cu^2+^ (µM)	5.8 ± 1.2	-	-	-
La-DDC + Cu^2+^ (µM)	-	-	89.7 ± 12.6	-
Ga-DDC + Cu^2+^ (µM)	-	-	17.7 ± 8.2	-
Ma-DDC + Cu^2+^ (µM)	-	-	78.6 ± 12.6	-

* (-) Denotes that certain IC_50_ values were not determined as this study represents an initial screening phase aimed at establishing proof of concept for anticancer activity across a range of saccharide-linked DDC derivatives in different cancer cell lines.

## Data Availability

Data is contained within the article and Appendix A.

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
