# Peer review of "Sugar-Linked Diethyldithiocarbamate Derivatives: A Novel Class of Anticancer Agents"

_ijms, 2025, doi:10.3390/ijms26125589_

Round 1
Reviewer 1 Report
Comments and Suggestions for Authors
Comments, suggestions
The authors present a well-focused study about saccharide-linked diethyldithiocarbamates (DDC), aiming to improve the properties of DDC as a potential anticancer agent.
To better understand the context and the relevance of the work, the cancer cell selectivity and the controlled release of DDC from prodrugs could be discussed in more detail in the introduction, with additional relevant references, addressing these aspects. For the discussion of the anticancer effect of the parent dug (disulfiram), a systematic review could have been cited to give a better overview.
Figure 2 and 3 do not show the data for all the novel derivatives, this would merit further explanation. Data for the missing derivatives is not presented in the SI either.
Please check the legend of Figure 3: G-DDC was also tested in DMSO:water 50:50? If yes, was it due to solubility reasons (it seems contradictory with the increase solubility expected from the prodrugs)?
In Table 1, where no values are indicated: they were not determined or were above a certain value? Please clarify this issue. Experiment with Zn2+ was run only with DG-DDC?
Method 4.2: what is supernatant B?
Method 4.3: what was the concentration of zinc in the medium for the experiment in the presence of zinc?
Method 4.4: time intervals do not refer to withdrawing samples instead of adding G-DDC to CuCl2?
Methods 4.2, 4.4 are presented with G-DDC – are they general methods used also for the other derivatives? If yes, please reword them as general methods.
Were cancer cell selectivity tests considered (i.e. selective cytotoxicity vs normal cells)?
Were solubility measurements considered? As this is one of the rationale for preparing these type of prodrugs, quantitative data should be provided.
Are all products (a-g) already described in the literature? Even in that case, some basic characterisation data could be provided in the SI (i.e. 1H NMR). For novel, not yet described products (e.g. XY-DDC) full synthetic and characterisation data should be provided (NMR, MS).
Technical issues, typos:
-line 15, 36, Figure 1 legend, line 88, 89, abbreviations: N,N in italics
-Table 1: in column 1 please indicate the charges for the ions (i.e. Cu2+, Zn2+)
-line 195: CO2
-in the References please check carefully all the references, some authors' list contain misplaced abbreviations and some sources do not seem fully correct
-author contributions: the instruction section can be removed (i.e. „For research articles with several authors, a short paragraph specifying their individual contributions must be provided. The following statements should be used”)
Overall, despite the valuable results presented, the scope of the study is limited both in terms of chemistry and biology. The introduction and the presentation of the data could be improved, as well as the description of the methods. Further characterisation data should be provided to give a more comprehensive picture. In the present form, the manuscript might not meet the criteria of a full IJMS paper.
Author Response
We would like to thank the reviewers for their insightful comments and constructive suggestions. We have carefully revised the manuscript and addressed all points raised. Below, we provide a detailed point-by-point response, with reviewer comments included in italics.
- Reviewer Comment:
The authors present a well-focused study about saccharide-linked diethyldithiocarbamates (DDC), aiming to improve the properties of DDC as a potential anticancer agent.
Response:
We thank the reviewer for their kind and encouraging comment recognising the focus and relevance of our study. This is greatly appreciated by all authors and reinforces our motivation to further develop this platform.
- Reviewer Comment:
To better understand the context and the relevance of the work, the cancer cell selectivity and the controlled release of DDC from prodrugs could be discussed in more detail in the introduction, with additional relevant references, addressing these aspects. For the discussion of the anticancer effect of the parent dug (disulfiram), a systematic review could have been cited to give a better overview.
Response:
We thank the reviewer for this helpful suggestion. In response, we have revised the Introduction to provide a clearer explanation of the rationale for designing saccharide-linked DDC prodrugs with tumour-selective activation. We have now explicitly discussed the concept of cancer cell selectivity, particularly how the prodrugs are designed to exploit elevated copper levels and metabolic features in the tumour microenvironment to achieve controlled activation. We have also highlighted the importance of prodrug stability in systemic circulation and the strategic shielding of the thiol group via thioglycosidic bonds.
Furthermore, we have added reference to a recent review that summarises the anticancer mechanisms of disulfiram and its copper-dependent limitations in clinical application [8]. This citation provides the broader overview the reviewer requested, while supporting our rationale for developing more stable, tumour-targeted delivery approaches.
- Reviewer Comment:
Figure 2 and 3 do not show the data for all the novel derivatives, this would merit further explanation. Data for the missing derivatives is not presented in the SI either.
Response:
We thank the reviewer for this observation. In the revised manuscript, we have clarified that all studied derivatives demonstrated similar stability in human serum and comparable reactivity with copper ions. However, to avoid redundancy and maintain clarity, we selected a representative subset of compounds—chosen based on preliminary activity and structural diversity—for detailed presentation in Figures 2 and 3. These examples illustrate the general behaviour of the saccharide-linked DDC series in terms of thioglycosidic stability and Cu²⁺-mediated activation. A statement reflecting this rationale has now been added to the end of the paragraph discussing these figures. Broader cytotoxicity screening is ongoing to prioritise compounds for further biological evaluation
Reviewer Comment:
Please check the legend of Figure 3: G-DDC was also tested in DMSO:water 50:50? If yes, was it due to solubility reasons (it seems contradictory with the increase solubility expected from the prodrugs)?
Response:
We apologise for the confusion. All saccharide-linked DDC derivatives, including G-DDC, were tested in aqueous solution only. DSF was the only compound tested in a 50:50 DMSO:water mixture due to its poor aqueous solubility, which significantly slowed its reactivity in water alone. This has now been clarified in the legend of Figure 3 and in figure 3 to avoid any ambiguity.
- Reviewer Comment:
In Table 1, where no values are indicated: they were not determined or were above a certain value? Please clarify this issue. Experiment with Zn2+ was run only with DG-DDC?
Response:
We thank the reviewer for this helpful query. For entries in Table 1 where no IC₅₀ values are shown, the values were not determined. We have now clarified this with appropriate footnotes in the table. Regarding the Zn²⁺ experiment, it was conducted only with DG-DDC as a representative compound to explore the effect of competitive metal ions on cytotoxicity.
Reviewer Comment:
Method 4.2: what is supernatant B?
Response:
We thank the reviewer for highlighting this point. Supernatant B refers to the methanol extract obtained after resuspending and centrifuging the pellet resulting from the initial aqueous extraction step. To avoid confusion, we have now amended the text in Method 4.2 to clarify this process.
- Reviewer Comment:
Method 4.3: what was the concentration of zinc in the medium for the experiment in the presence of zinc?
Response:
We thank the reviewer for pointing this out. The concentration of Zn²⁺ used in the experiment was 10 µM. This has now been clearly stated in the revised text of Method 4.3.
- Reviewer Comment:
Method 4.4: time intervals do not refer to withdrawing samples instead of adding G-DDC to CuCl2?
Response:
We thank the reviewer for this helpful clarification. The time intervals indeed refer to the withdrawal of aliquots following the initiation of the reaction, not the timing of G-DDC addition. We have revised Method 4.4 to explicitly reflect this sequence and avoid ambiguity.
- Reviewer Comment:
Methods 4.2, 4.4 are presented with G-DDC – are they general methods used also for the other derivatives? If yes, please reword them as general methods.
Response:
We thank the reviewer for this useful observation. The methods described in Sections 4.2 and 4.4 were indeed applied to all synthesised saccharide-linked DDC derivatives using equivalent conditions. We have revised the wording of these sections to reflect their general applicability rather than limiting the description to G-DDC alone.
- Reviewer Comment:
Were cancer cell selectivity tests considered (i.e. selective cytotoxicity vs normal cells)?
Response:
We thank the reviewer for this important suggestion. While the current study focused on demonstrating copper-mediated activation and anticancer potential across a panel of cancer cell lines, we agree that assessing tumour selectivity is a valuable next step. We plan to include comparative cytotoxicity studies using non-cancerous cell lines in future work to explore the selectivity and therapeutic window of the saccharide-linked DDC derivatives. This has now been noted in the Discussion section as part of the intended scope of further investigations.
- Reviewer Comment:
Were solubility measurements considered? As this is one of the rationale for preparing these type of prodrugs, quantitative data should be provided.
Response:
We thank the reviewer for this relevant observation. All saccharide-linked DDC derivatives demonstrated high aqueous solubility, with solubility exceeding 100 mg/mL in water. As a result, quantitative determination was challenging using standard analytical methods. However, this high solubility aligns with the rationale for the prodrug design and is now mentioned in the Discussion section of the manuscript as a practical advantage for formulation and delivery.
- Reviewer Comment:
Are all products (a-g) already described in the literature? Even in that case, some basic characterisation data could be provided in the SI (i.e. 1H NMR). For novel, not yet described products (e.g. XY-DDC) full synthetic and characterisation data should be provided (NMR, MS).
Response:
We thank the reviewer for this suggestion. All previously reported compounds in the study were characterised by H1 NMR and MS. However, we believe that including full characterisation data in the Supplementary Information is most appropriate for novel compounds. Accordingly, we have provided ¹H NMR and MS for XY-DDC in the revised Supplementary Information, as it has not been previously reported.
Reviewer Comment:
Technical issues, typos:
-line 15, 36, Figure 1 legend, line 88, 89, abbreviations: N,N in italics
-Table 1: in column 1 please indicate the charges for the ions (i.e. Cu2+, Zn2+)
-line 195: CO2
-in the References please check carefully all the references, some authors' list contain misplaced abbreviations and some sources do not seem fully correct
-author contributions: the instruction section can be removed
Response:
We thank the reviewer for the detailed and constructive comments. We have carefully reviewed and corrected all technical and formatting issues noted.
- Reviewer Comment:
Overall, despite the valuable results presented, the scope of the study is limited both in terms of chemistry and biology. The introduction and the presentation of the data could be improved, as well as the description of the methods. Further characterisation data should be provided to give a more comprehensive picture. In the present form, the manuscript might not meet the criteria of a full IJMS paper.
Response:
We thank the reviewer for their thoughtful evaluation and acknowledge the points raised regarding the scope and clarity of the manuscript. In response, we have made several substantive improvements:
- The Introduction has been revised to better articulate the relevance of cancer cell selectivity and controlled prodrug activation, with additional references to support the rationale and highlight recent findings.
- We have clarified the methods, particularly Sections 4.2 and 4.4, to reflect their general applicability to all derivatives rather than G-DDC alone.
- We have provided full NMR and MS characterisation data in the Supplementary Information for novel compounds, such as XY-DDC, in line with the reviewer’s suggestion.
- Minor technical and formatting issues throughout the manuscript and reference list have been addressed.
While this work represents a focused proof-of-concept study, we believe that the mechanistic insights, compound design strategy, and biological findings provide a meaningful contribution to the field and now meet the expectations for a full paper in IJMS.
Reviewer 2 Report
Comments and Suggestions for Authors
The work presented by M. Najlah et al. shows an interesting study on the application of DDC metabolite derivatives conjugated to sugars to increase the potential therapeutic application against cancer. The work is very well presented and explained, but some things are missing to complete the work:
- Characterization of the compounds formed, in order to demonstrate that the conjugation has been done correctly.
- The in vitro studies are quite good as it compares the activity of all compounds in four different cancer lines. However, to complete the study well, it is advisable to perform the assays in at least one healthy cell line.
- Be careful with superscripts (e.g. in CO2)
Author Response
The work presented by M. Najlah et al. shows an interesting study on the application of DDC metabolite derivatives conjugated to sugars to increase the potential therapeutic application against cancer. The work is very well presented and explained, but some things are missing to complete the work:
Response:
We sincerely thank the reviewer for the positive evaluation of our study and for acknowledging the clarity and relevance of the work. We have carefully considered the suggestions provided and have taken steps to address all points raised. We believe these revisions improve the completeness and impact of the manuscript.
- Reviewer Comment:
Characterization of the compounds formed, in order to demonstrate that the conjugation has been done correctly.
Author Response:
We thank the reviewer for this important point. All synthesised saccharide-linked DDC derivatives were characterised by ¹H NMR and mass spectrometry. Full characterisation data for the novel compound XY-DDC, which has not been previously reported in the literature, have now been included in the Supplementary Information to confirm the structure and demonstrate successful conjugation. Previously reported compounds were characterised and verified according to established literature procedures.
- Reviewer Comment:
The in vitro studies are quite good as it compares the activity of all compounds in four different cancer lines. However, to complete the study well, it is advisable to perform the assays in at least one healthy cell line.
Author Response:
We appreciate the reviewer’s constructive suggestion. While the current study focused on evaluating copper-mediated cytotoxicity in cancer cell lines, we agree that including healthy cell lines is essential for assessing tumour selectivity. This has now been noted in the conclusion section as part of our planned future work, where comparative cytotoxicity in non-cancerous cells will be undertaken to further define the therapeutic window of the compounds.
- Reviewer Comment:
Be careful with superscripts (e.g. in CO₂).
Author Response:
We thank the reviewer for noting this formatting issue. We have reviewed the manuscript thoroughly and corrected all formatting inconsistencies related to superscripts, including CO₂ and ion charges (e.g. Cu²⁺, Zn²⁺).
Round 2
Reviewer 1 Report
Comments and Suggestions for Authors
In the revised version of the manuscript several corrections were integrated, that improve its overall quality. Particulary the introduction and the methods section were revised and several minor technical or formatting issues were fixed.
Comments:
-despite the authors' claim about adding additional references, no new reference was added, the references are only given in a slightly different order
-line 106,107: N,N in italics
-Table 1: a clarifying footnote was added, although it does not discuss the reason for not determining the given values. However, for XY-DDC + Cu2+ there are neither values, nor - signs, except for the first cell.
-please provide a better resolution version of Figure 4
-line 200: and characterised
-for the synthesis part, it would be useful for readers to note which compounds are already reported in the literature and provide the reference (if it is different from reference 11)
-line 202: please reword - "A 0.5 ml sample of 1 mmol of the saccharide-DDC derivative was preheated at"
-SI: please check Figure 3S legend - 225 µM 2-deoxy-glycosyl diethyldithiocarbamate
Overall, despite the valuable results presented, the scope of the study is limited both in terms of chemistry and biology. In the present form, the manuscript might not meet the criteria of a full IJMS paper and can be considered as a brief report.
Author Response
We sincerely thank the reviewer for their continued careful assessment and constructive comments, which have helped us further improve the quality and clarity of our manuscript. Below, we provide detailed responses to each of the reviewer’s comments, outlining the revisions made in the latest version of the manuscript and Supplementary Information.
Reviewer Comment:
Despite the authors’ claim about adding additional references, no new reference was added; the references are only given in a slightly different order.
Author Response:
We apologise for the misunderstanding in our previous response. We had initially interpreted the comment as a request to re-cite an existing reference rather than to add new ones. In this revised version, we have now added two new references, including a systematic review on disulfiram’s anticancer effects. These references are cited in the Introduction to strengthen the rationale for developing saccharide-linked DDC prodrugs.
Reviewer Comment:
Line 106,107: N,N in italics.
Author Response:
Corrected as requested.
Reviewer Comment:
Table 1: a clarifying footnote was added, although it does not discuss the reason for not determining the given values. However, for XY-DDC + Cu²⁺ there are neither values, nor ‘–’ signs, except for the first cell.
Author Response:
We thank the reviewer for this helpful observation. We have now updated the footnote in Table 1 to clarify that certain IC₅₀ values were not determined as this study represents an initial screening phase aimed at establishing proof of concept for anticancer activity across a range of saccharide-linked DDC derivatives in different cancer cell lines. The focus was on identifying representative activity rather than generating exhaustive IC₅₀ data for all compounds. For XY-DDC + Cu²⁺, we added ‘–’ symbols as data were not determined, ensuring the table is consistent and fully annotated.
Reviewer Comment:
Please provide a better resolution version of Figure 4.
Author Response:
A higher-resolution version of Figure 4 has been provided in the revised manuscript to enhance visual clarity. In addition, Figures 2 and 3 have also been updated to improve consistency and presentation quality.
Reviewer Comment:
Line 200: and characterised.
Author Response:
The sentence has been corrected as suggested and now reads:
“The compounds were synthesised and characterised…”
Reviewer Comment:
For the synthesis part, it would be useful for readers to note which compounds are already reported in the literature and provide the reference (if it is different from reference 11).
Author Response:
We thank the reviewer for this helpful suggestion. In the revised manuscript, we have now clarified that all saccharide derivatives with diethyl and/or dimethyl substitution have been previously reported in the literature and are now properly referenced (now number 13).
As the synthesis involves a straightforward one-step thioglycosylation, we focused this work on demonstrating the biological concept and copper-activation mechanism rather than synthetic novelty. Nonetheless, if the reviewer feels it would strengthen the manuscript, we are happy to include full NMR spectra for all derivatives in the Supplementary Information.
Reviewer Comment:
Line 202: please reword - “A 0.5 ml sample of 1 mmol of the saccharide-DDC derivative was preheated at”
Author Response:
We thank the reviewer for this helpful suggestion. The sentence has been revised for clarity and to follow standard scientific phrasing. It now reads:
“A 0.5 mL aqueous solution of the saccharide-DDC derivative (1 mmol) was preheated at 37 °C.”
Reviewer Comment:
SI: please check Figure 3S legend - 225 µM 2-deoxy-glycosyl diethyldithiocarbamate.
Author Response:
We thank the reviewer for their careful attention to detail. The figure legend for Figure 3S in the Supplementary Information has now been corrected to accurately read: “25 µM 2-deoxy-glycosyl diethyldithiocarbamate
Reviewer 2 Report
Comments and Suggestions for Authors
I believe that, although they have not performed the additional biological experiments I suggested, the work may be suitable for publication in this journal.
Author Response
Thank you for your supportive comment and constructive feedback.